# A Self Validation Network for Object-Level Human Attention Estimation

**Zehua Zhang,**[1] **Chen Yu,**[2] **David Crandall**[1]
[1]Luddy School of Informatics, Computing, and Engineering
[2]Department of Psychological and Brain Sciences
Indiana University Bloomington
{zehzhang, chenyu, djcran}@indiana.edu

## Abstract

Due to the foveated nature of the human vision system, people can focus their visual attention on only a small region of their visual field at a time, which usually contains a single object. Estimating this object of attention in first-person (egocentric) videos is useful for many human-centered real-world applications such as augmented reality and driver assistance systems. A straightforward solution for this problem is to first estimate the gaze with a traditional gaze estimator and generate object candidates from an off-the-shelf object detector, and then pick the object that the estimated gaze falls in. However, such an approach can fail because it addresses the *where* and the *what* problems separately, despite that they are highly related, chicken-and-egg problems. In this paper, we propose a novel unified model that incorporates both spatial and temporal evidence in identifying as well as locating the attended object in first-person videos. It introduces a novel Self Validation Module that enforces and leverages consistency of the *where* and the *what* concepts. We evaluate on two public datasets, demonstrating that the Self Validation Module significantly benefits both training and testing and that our model outperforms the state-of-the-art.

## 1 Introduction

Humans can focus their visual attention on only a small part of their surroundings at any moment, and thus have to choose what to pay attention to in real time [43]. Driven by the tasks and intentions we have in mind, we manage attention with our foveated visual system by adjusting our head pose and our gaze point in order to focus on the most relevant object in the environment at any moment in time [8, 17, 29, 47, 62].

This close relationship between intention, attention, and semantic objects has inspired a variety of work in computer vision, including image classification [26], object detection [27, 46, 50, 52], action recognition [4, 36, 42, 48], action prediction [53], video summarization [30], visual search modeling [51], and irrelevant frame removal [38], in which the attended object estimation serves as auxiliary information. Despite being a key component of these papers, how to identify and locate the important object is seldom studied explicitly. This problem in and of itself is of broad potential use in real-world applications such as driver assistance systems and intelligent human-like robots.

In this paper, we discuss how to identify and locate the attended object in first-person videos. Recorded by head-mounted cameras along with eye trackers, first-person videos capture an approximation of what people see in their fields of view as they go about their lives, yielding interesting data for studying real-time human attention. In contrast to gaze studies of static images or pre-recorded videos, first-person video is unique in that there is exactly one correct point of attention in each frame, as a camera wearer can only gaze at one point at a time. Accordingly, one and only one gazed object

exists for each frame, reflecting the camera wearer's real-time attention and intention. We will use the term ***object of interest*** to refer to the attended object in our later discussion.

Some recent work [22, 66, 68] has discussed estimating probability maps of ego-attention or predicting gaze points in egocentric videos. However, people think not in terms of points in their field of view, but in terms of the *objects* that they are attending to. Of course, the object of interest could be obtained by first estimating the gaze with the gaze estimator and generating object candidates from an off-the-shelf object detector, and then picking the object that the estimated gaze falls in. Because this bottom-up approach estimates *where* and *what* separately, it could be doomed to fail if the eye gaze prediction is slightly inaccurate, such as falling between two objects or in the intersection of multiple object bounding boxes (Figure 1). To assure consistency, one may think of performing anchor-level attention estimation and directly predicting the attended box by modifying existing object detectors. Class can be either predicted simultaneously with the anchor-level attention estimation using the same set of features, as in SSD [40], or afterwards using the

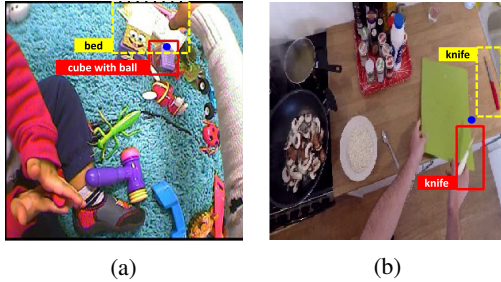

(a)                              (b)

Figure 1: *Among the many objects appearing in an egocentric video frame of a person's field of view, we want to identify and locate the object to which the person is visually attending.* Combining traditional eye gaze estimators and existing object detectors can fail when eye gaze prediction (blue dot) is slightly incorrect, such as when (a) it falls in the intersection of two object bounding boxes or (b) it lies between two bounding boxes sharing the same class. Red boxes shown actual attended object according to ground truth gaze and yellow dashed boxes show incorrect predictions.

features pooled within the attended box, as in Faster-RCNN [49]. Either way, these methods still do not yield satisfying performance, as we will show in Sec. 4.2, because they lack the ability to leverage the consistency to refine the results.

We propose to identify and locate the object of interest by *jointly* estimating *where* it is within the frame as well as recognizing *what* its identity is. In particular, we propose a novel model — which we cheekily call Mindreader Net or Mr. Net — to jointly solve the problem. Our model incorporates both spatial evidence within frames and temporal evidence across frames, in a network architecture (which we call the Cogged Spatial-Temporal Module) with separate spatial and temporal branches to avoid feature entanglement.

A key feature of our model is that it explicitly enforces and leverages a simple but extremely useful constraint: our estimate of *what* is being attended should be located in exactly the position of *where* we estimate the attention to be. This Self Validation Module first computes similarities between the global object of interest class prediction vector and each local anchor box class prediction vector as the attention validation score to update the anchor attention score prediction, and then, with the updated anchor attention score, we select the attended anchor and use its corresponding class prediction score to update the global object of interest class prediction. With global context originally incorporated by extracting features from the whole clip using 3D convolution, the Self Validation Module helps the network focus on the local context in a spatially-local anchor box and a temporally-local frame.

We evaluate the approach on two existing first-person video datasets that include attended object ground truth annotations. We show our approach outperforms baselines, and that our Self Validation Module not only improves performance by refining the outputs with visual consistency during testing, but also it helps bridge multiple components together during training to guide the model to learn a highly meaningful latent representation. More information is available at http://vision.soic.indiana.edu/mindreader/.

## 2   Related Work

Compared with many efforts to understand human attention by modeling eye gaze [2, 7, 16, 20–22, 24, 34, 35, 45, 59, 60, 64, 66, 68] or saliency [19, 25, 31–33, 39, 55, 67, 69], there are relatively few papers that detect object-level attention. Lee *et al.* [30] address video summarization with hand-crafted features to detect important people and objects, while object-level reasoning plays a key

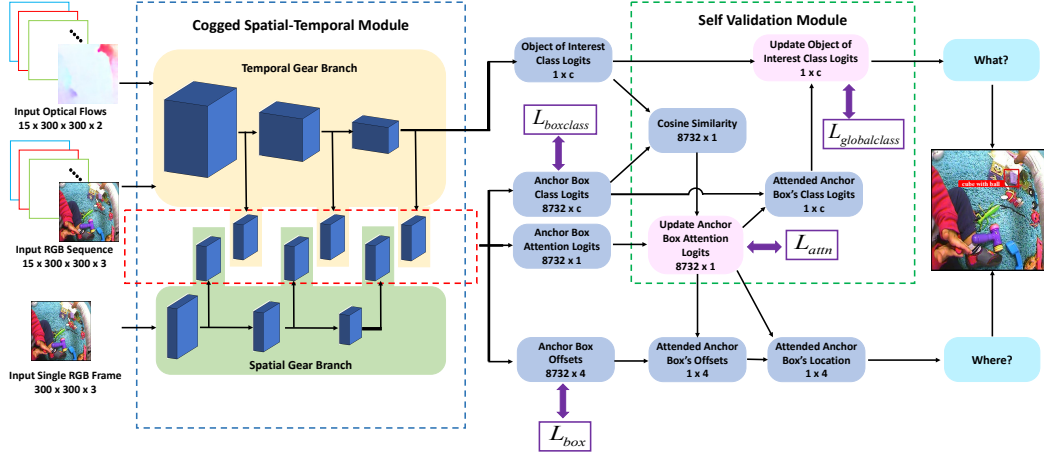

Figure 2: *The architecture of our proposed Mindreader Net.* Numbers indicate output size of each component (where $c$ is the number of object classes). Softmax is applied before computing the losses on global classification $L_{globalclass}$, anchor box classification $L_{boxclass}$, and attention $L_{attn}$ (which is first flattened to be 8732-d). Please refer to supplementary materials for details about the Cogged Spatial-Temporal Module.

role in Baradel *et al.*'s work on understanding videos through interactions of important objects [4]. In the particular case of egocentric video, Pirsiavash and Ramanan [48] and Ma *et al.* [42] detect objects in hands as a proxy for attended objects to help action recognition. However, eye gaze usually precedes hand motion and thus objects in hand are not always those being visually attended (Fig. 1a). Shen *et al.* [53] combine eye gaze ground truth and detected object bounding boxes to extract attended object information for future action prediction. EgoNet [5], among the first papers to focus on important object detection in first-person videos, combines visual appearance and 3D layout information to generate probability maps of object importance. Multiple objects can be detected in a single frame, making their results more similar to saliency than human attention in egocentric videos.

Perhaps the most related work to ours is Bertasius *et al.*'s Visual-Spatial Network (VSN) [6], which proposes an unsupervised method for important object detection in first-person videos that incorporates the idea of consistency between the *where* and *what* concepts to facilitate learning. However, VSN requires a much more complicated training strategy of switching the cascade order of the two pathways multiple times, whereas we present a unified framework that can be learned end-to-end.

# 3    Our approach

Given a video captured with a head-mounted camera, our goal is to detect the object that is visually attended in each frame. This is challenging because egocentric videos can be highly cluttered, with many competing objects vying for attention. We thus incorporate temporal cues that consider multiple frames at a time. We first consider performing detection for the middle frame of a short input sequence (as in [42]), and then further develop it to work online (considering only past information) by performing detection on the last frame. Our novel model consists of two main parts (Figure 2), which we call the Cogged Spatial-Temporal Module and the Self Validation Module.

## 3.1    Cogged Spatial-Temporal Module

The Cogged Spatial-Temporal Module consists of a spatial and a temporal branch. The "cogs" refer to the way that the outputs of each layer of the two branches are combined together, reminiscent of the interlocking cogs of two gears (Figure 2). Please see supplementary material for more details.

**The Spatial Gear Branch,** inspired by SSD300 [40], takes a single frame $I_t$ of size $h \times w$ and performs spatial prediction of local anchor box offsets and anchor box classes. It is expected to work as an object detector, although we only have ground truth for the objects of interest to train it, so we do not add an extra background class as in [40], and only compute losses for the spatial-based tasks on the matched positive anchors. We use atrous [10, 65] VGG16 [54] as the backbone and follow a similar anchor box setting as [40]. We also apply the same multi-anchor matching strategy.

With the spatial branch, we obtain anchor box offset predictions $O \in R^{a \times 4}$ and class predictions $C_{box} \in R^{a \times c}$, where $a$ is the number of anchor boxes and $c$ is the number of classes in our problem. Following SSD300 [40], we have $a = 8732$, $h = 300$, and $w = 300$.

**The Temporal Gear Branch** takes $N$ continuous RGB frames $I_{t-\frac{N-1}{2},t+\frac{N-1}{2}}$ as well as $N$ corresponding optical flow fields $F_{t-\frac{N-1}{2},t+\frac{N-1}{2}}$, both of spatial resolution $h \times w$ (with $N = 15$, set empirically). We use Inception-V1 [58] I3D [9] as the backbone of our temporal branch. With aggregated global features from 3D convolution, we obtain global object of interest class predictions $C_{global} \in R^{1 \times c}$ and anchor box attention predictions $A \in R^{a \times 1}$. We match the ground truth box only to the anchor with the greatest overlap (intersection over union). The matching strategy is empirical and discussed in Section 4.3.

## 3.2 Self Validation Module

The Self Validation Module connects the above branches and delivers global and local context between the two branches at both spatial (*e.g.*, whole frame versus an anchor box) and temporal (*e.g.*, whole sequence versus a single frame) levels. It incorporates the constraint on consistency between where and what by embedding a double validation mechanism: what→where and where→what.

**What→where.** With the outputs of the Cogged Spatial-Temporal Module, we compute the cosine similarities between the global class prediction $C_{global}$ and the class prediction for each anchor box, $C_{box_i}$, yielding an attention validation score for each box $i$,

$$V_{attn}^i = \frac{C_{global}C_{box_i}^T}{||C_{global}|| \times ||C_{box_i}||}. \tag{1}$$

Then the attention validation vector $V_{attn} = [V_{attn}^1, V_{attn}^2, ..., V_{attn}^a] \in R^{a \times 1}$ is used to update the anchor box attention scores $A$ by element-wise summation, $A' = A + V_{attn}$. Since $-1 \leq V_{attn}^i \leq 1$, we make the optimization easier by rescaling each $A_i$ to the range $[-1, 1]$,

$$A' = R(A) + V_{attn} = \frac{A - (max(A) + min(A))/2}{max(A) - (max(A) + min(A))/2} + V_{attn}, \tag{2}$$

where $max()$ and $min()$ are element-wise vector operations.

**Where→what.** Intuitively, obtaining the attended anchor box index $m$ is a simple matter of computing $m = argmax(A')$, and the class validation score is simply $V_{class} = C_{box_m}$. Similarly, after rescaling, we take an element-wise summation, $V_{class}$ and $C_{global}$, to update the global object of interest class prediction ($R(\cdot)$ in Equation 2), $C'_{global} = R(C_{global}) + R(V_{class})$. However, the hard argmax is not differentiable, and thus gradients are not able to backpropagate properly during training. We thus use soft argmax. Softmax is applied to the updated anchor box attention score $A'$ to produce a weighting vector $\widetilde{A}'$ for class validation score estimation,

$$\hat{V}_{class} = \sum_{i=1}^{a} \widetilde{A}_i' C_{box_i}, \qquad \text{with } \widetilde{A}_i' = \frac{e^{A_i'}}{\sum_{j=1}^{a} e^{A_j'}} \tag{3}$$

Now we replace $V_{class}$ with $\hat{V}_{class}$ to update $C_{global}$, $C'_{global} = R(C_{global}) + R(\hat{V}_{class})$.

This soft what→where validation is closely related to the soft attention mechanism widely used in many recent papers [3, 11, 41, 56, 61, 63]. While soft attention learns the mapping itself inside the model, we explicitly incorporate the coherence of the where and what concepts into our model to self-validate the output during both training and testing. In contrast to soft attention which describes relationships between *e.g.* words, graph nodes, *etc.*, this self-validation mechanism naturally mirrors the visual consistency of our foveated vision system.

## 3.3 Implementation and training details

We implemented our model with Keras [12] and Tensorflow [1]. A batch normalization layer [23] is inserted after each layer in both spatial and temporal backbones, and momentum for batch normalization is 0.8. Batch normalization is not used in the four prediction heads. We found pretraining the spatial branch helps the model converge faster. No extra data is introduced as we

still only use the labels of the objects of interest for pretraining. VGG16 [54] is initialized with weights pretrained on ImageNet [14]. We use Sun *et al.*'s method [44, 57] to extract optical flow and follow [9] to truncate the maps to $[-20, 20]$ and then rescale them to $[-1, 1]$. The RGB input to the Temporal Gear Branch is rescaled to $[-1, 1]$ [9], while for the Spatial Gear Branch the RGB input is normalized to have 0 mean and the channels are permuted to BGR.

When training the whole model, the spatial branch is initialized with the pretrained weights from above. The I3D backbone is initialized with weights pretrained on Kinetics [28] and ImageNet [14], while other parts are randomly initialized. We use stochastic gradient descent with learning rate 0.03, momentum 0.9, decay 0.0001, and $L2$ regularizer $5e^{-5}$. The loss function consists of four parts: global classification $L_{globalclass}$, attention $L_{attn}$, anchor box classification $L_{boxclass}$, and box regression $L_{box}$,

$$L_{total} = \alpha L_{globalclass} + \beta L_{attn} + \frac{1}{N_{pos}}(\gamma L_{boxclass} + L_{box}), \qquad (4)$$

where we empirically set $\alpha = \beta = \gamma = 1$, and $N_{pos}$ is the total number of matched anchors for training the anchor box class predictor and anchor box offset predictor. $L_{globalclass}$ and $L_{attn}$ apply cross entropy loss, computed on the updated predictions of object of interest class and anchor box attention. $L_{boxclass}$ is the total cross entropy loss and $L_{box}$ is the total box regression loss over only all the matched anchors. The box regression loss follows [40, 49] and we refer readers there for details. Our full model has $64M$ trainable parameters, while the Self Validation Module contains no parameters, making it very flexible so that it can be added to training or testing anytime. It is even possible to stack multiple Self Validation Modules or use only half of it.

During testing, the anchor with the highest anchor box attention score $A'_i$ is selected as the attended anchor. The corresponding anchor box offset prediction $O_i$ indicates where the object of interest is, while the argmax of the global object of interest class score $C'_{global}$ gives its class.

## 4   Experiments

We evaluate our model on identifying attended objects in two first-person datasets collected in very different contexts: child and adult toy play, and adults in kitchens.

**ATT [68]** (Adult-Toddler Toy play) consists of first-person videos from head-mounted cameras of parents and toddlers playing with 24 toys in a simulated home environment. The dataset consists of 20 synchornized video pairs (child head cameras and parent head cameras), although we only use the parent videos. The object being attended is determined using gaze tracking. We randomly select 90% of the samples in each object class for training and use the remaining 10% for testing, resulting in about $17,000$ training and $1,900$ testing samples, each with 15 continuous frames. We do not restrict the object of interest to remain the same in each sample sequence and only use the label of the object of interest for training.

**Epic-Kitchen Dataset [13]** contains 55 hours of first-person video from 32 participants in their own kitchens. The dataset includes anntoations on the "active" objects related to the person's current action. We use this as a proxy for attended object by we selecting only frames containing one active object and assuming that they are attended. Object classes with fewer than 1000 samples are also excluded, resulting in 53 classes. We randomly select 90% of samples for training, yielding about $120,000$ training and $13,000$ testing samples.

For evaluation, we report accuracy — number of correct predictions over the number of samples. A prediction is considered correct if it has both (a) the correct class prediction and (b) an IoU between the estimated and the ground truth boxes above a threshold. Similar to [37], we report accuracies at IOU thresholds of $0.5$ and $0.75$, as well as a mean accuracy $mAcc$ computed by averaging accuracies at 10 IOU thresholds evenly distributed from 0.5 to 0.95. Accuracy thus measures ability to correctly predict both what and where is being attended.

### 4.1   Baselines

We evaluate against several strong baselines. **Gaze + GT bounding box,** inspired by Li *et al.* [35], applies Zhang *et al.*'s gaze prediction method [68] (since it has state-of-the-art performance on the *ATT*) and directly uses ground truth object bounding boxes. This is equivalent to having a perfect

| Method | $Acc_{0.5}$ ↑ | $Acc_{0.75}$ ↑ | $mAcc$ ↑ |
|---|---|---|---|
| Our Mr. Net | **74.27** | **46.78** | **44.78** |
| Gaze [68] + GT Box + Hit | 25.26 | 25.26 | 25.26 |
| Gaze [68] + GT Box + Closest | 35.86 | 35.86 | 35.86 |
| I3D [9]-based SSD [40] | **70.11** | 42.10 | 40.85 |
| Cascade Model | 66.97 | **45.10** | 41.93 |
| OIH Detectors + WH Classifier | 37.16 | 37.16 | 37.16 |
| Left Handed Model | 38.31 | 38.31 | 38.31 |
| Right Handed Model | 39.00 | 39.00 | 39.00 |
| OIH GT + WH Classifier | 40.83 | 40.83 | 40.83 |
| Either Handed Model | 42.94 | 42.94 | **42.94** |
| Center GT Box | 23.97 | 23.97 | 23.97 |

Table 1: *Accuracy of our method compared to others,* on the ATT dataset. OIH represents Object-in-Hand, while WH means Which-Hand.

| | Self validation? | | | | |
|---|---|---|---|---|---|
| Streams | Training | Testing | $Acc_{0.5}$ ↑ | $Acc_{0.75}$ ↑ | $mAcc$ ↑ |
| Two | yes | yes | **74.27** | **46.78** | **44.78** |
| Two | yes | half | — | — | 43.88 |
| Two | yes | no | 68.19 | 42.83 | 41.18 |
| Two | no | yes | 67.18 | 40.06 | 39.48 |
| Two | no | half | — | — | 37.87 |
| Two | no | no | 62.33 | 38.31 | 37.18 |
| RGB | yes | yes | 74.59 | 43.15 | 42.48 |
| Flow | yes | yes | 64.30 | 38.63 | 37.60 |
| Flow | no | yes | — | — | 25.10 |
| Flow | no | no | — | — | 18.40 |

Table 2: *Ablation results.* Testing with half means that the model is tested with only what→where validation.

object detector (with $mAP = 100\%$), resulting in a very strong baseline. We use two different methods to match the predicted eye gaze to the object boxes: (1) **Hit:** only boxes in which the gaze falls in are considered matched, and if the estimated gaze point is within multiple boxes, the accuracy score is averaged by the number of matched boxes; and (2) **Closest:** the box whose center is the closest to the predicted gaze is considered to be matched. **I3D [9]-based SSD [40]** tries to overcome the discrepancy caused by solving the *where* and *what* problems separately by directly performing anchor-level attention estimation with an I3D [9]-backboned SSD [40]. The anchor box setting is similar to SSD300 [40]. For each anchor we predict an attention score, a class score, and box offsets. **Cascade model** contains a temporal branch with I3D backbone and a spatial branch with VGG16 backbone. From the temporal branch, the important anchor as well as its box offsets are predicted, and then features are pooled [18, 49] from the spatial branch for classification. **Object in hands + GT bounding box,** inspired by [15, 42, 48], tries to detect object of interest by detecting the object in hand. We use several variants; the "either handed model" is strongest, and uses both the ground truth object boxes and the ground truth label of the object in hands. When two hands hold different objects, the model always picks the one yielding higher accuracy, thus reflecting the best performance we can obtain with this baseline. Please refer to the supplementary materials for details of other variants. **Center GT box** uses the ground truth object boxes and labels to select the object closest to the frame center, inspired by the fact that people tend to adjust their head pose so that their gaze is near the center of their view [34].

## 4.2   Results on ATT dataset

Table 1 presents quantitative results of our Mindreader Net and baselines on the *ATT* dataset. Both enforcing and leveraging the visual consistency, our method even outperformed the either-handed model in terms of $mAcc$, which is built upon several strong *oracles* — a perfect object detector, two perfect object-in-hand detectors, and a perfect which-hand classifier. Other methods without perfect object detectors suffer from a rapid drop in $Acc$ as the IOU threshold becomes higher. For example, when the IOU threshold reaches 0.75, the either-handed model already has no obvious advantage compared with I3D-based SSD, and the Cascade model achieves a much higher score. When the threshold becomes 0.5, not only our Mindreader Net but also Cascade and I3D-based SSD outperform the either-handed model by a significant margin. Though the $Acc_{0.5}$ of the cascade model is lower than I3D-based SSD by about $3\%$, its $mAcc$ and $Acc_{0.75}$ are higher, suggesting bad box predictions with low IOU confuses the class head of the cascade model, but having a separate spatial branch to overcome feature entanglement improves the overall performance with higher-quality predictions.

We also observed that the Closest variant of the Gaze + GT Box model is about $40\%$ better than the Hit variant. This suggests that gaze prediction often misses the ground truth box a bit or may fall in the intersection of several bounding boxes, reflecting the discrepancy between the *where* and the *what* concepts in exiting eye gaze estimation algorithms.

Sample results of our model compared with other baselines are shown in Figure 3. Regular gaze prediction models fail in (c) & (d), supporting our hypothesis about the drawback of estimating where and what independently — the model is not robust to small errors in gaze estimation (recall the gaze-based baseline uses ground truth bounding boxes so failures must be caused by gaze estimation). In particular, the estimated gaze falls on 3 objects in (c), slightly closer to the center of the rabbit; In (d), eye gaze does not fall on any object. More unified models (I3D-based SSD, the cascade model,

and our model) thus achieve better performance. In (a) & (b), our model outperforms I3D-based SSD and Cascade. Because a Self Validation Module is applied to inject consistency, our Mr. Net performs better when many objects including the object of interest are close to each other.

Figure 4 illustrate how various parts of our model work. Image (a) shows the intermediate anchor attention score $A \in R^{a \times 1}$ from the temporal branch, visualized as the top 5 attended anchors with attention scores. These are anchor-level attention and no box offsets are predicted here. Image (b) shows visualizations of the predicted anchor offsets $O \in R^{a \times 4}$ and box class score $C_{box} \in R^{a \times c}$ from the spatial branch (only of the top 5 attended anchors). We do not have negative samples or a background class for training the spatial branch and thus there are some false positives. Image (c) combines output from both branches; this is also the final prediction of the model trained with the Self Validation Module but tested without it in the ablation studies in Section 4.3. The predicted class is obtained from $C_{global}$ and we combine $A$ and $O$ to get the location. Discrepancy happens in this example as the class prediction is correct but not the location. Image (d) shows prediction of our full model. By applying double self validation, the full model correctly predicts location and class.

Some failure cases of our model are shown in Figure 5: (a) heavy occlusion, (b) ambiguity of which held object is attended, (c) the model favors the object that is reached for, and (d) an extremely difficult case where parent's reach is occluded by an object held by the child.

## 4.3 Ablation studies

We conduct several ablation studies to evaluate the importance of the parts of our model.

**Hard argmax vs. soft argmax during testing.** The soft version of what→where is necessary for gradient backpropagation during training, but there is no such issue in testing. Our full model achieves $mAcc = 44.78\%$ when tested with hard argmax, versus $mAcc = 44.13\%$ when tested with soft argmax. When doing the same experiments with other model settings, we observed similar results.

**Self Validation Module.** To study the importance of the Self Validation Module, we conduct five experiments: (1) Train and test the model without the Self Validation Module; (2) Train the model without the Self Validation Module but test with only the what→where validation (the first step of Self Validation); (3) Train the model without Self Validation but test with it; (4) Train the model with Self Validation but test with only what→where validation; (5) Train the model with Self Validation but test without it. As shown in Table 2, the Self Validation Module yields consistent performance gain. If we train the model with Self Validation but remove it during testing, the remaining model still outperforms other models trained without the module. This implies that embedding the Self Validation Module during training helps learn a better model by bridging each component and providing guidance of how components are related to each other. Even when Self Validation is removed during testing, consistency is still maintained between the temporal and the spatial branches. Also, recall that when training the model with the Self Validation Module, the loss is computed based on the final output, and thus when we test the full model without Self Validation, the output is actually a latent representation in our full model. This suggests that our Self Validation Module encourages the model to learn a highly semantically-meaningful latent representation. Furthermore, the consistency injected by Self Validation helps prevent overfitting, while significant overfitting was observed without the Self Validation Module during training.

**Validation method for what→where.** We used element-wise summation for what→where validation. Another strategy is to treat $V_{attn}$ as an attention vector in which rescaling is unnecessary,

$$A'_i = A_i \cdot \widetilde{V}^i_{attn}, \qquad \text{with} \quad \widetilde{V}^i_{attn} = \frac{e^{V^i_{attn}}}{\sum_{j=1}^{a} e^{V^j_{attn}}}. \tag{5}$$

We repeated experiments using this technique and obtained $mAcc = 43.30\%$, a slight drop that may be because the double softmax inside the Self Validation Module increases optimization difficulty.

**Single stream versus two streams.** We conducted experiments to study the effect of each stream in our task. As Table 2 shows, a single optical flow stream performs much worse than single RGB or two-stream, indicating that object appearance is very important for problems related to object detection. However, it acheived acceptable results since the network can refer to the spatial branch for appearance information through the Self Validation Module. To test this, we removed the Self Validation Module from the single flow stream model during training. When testing this model

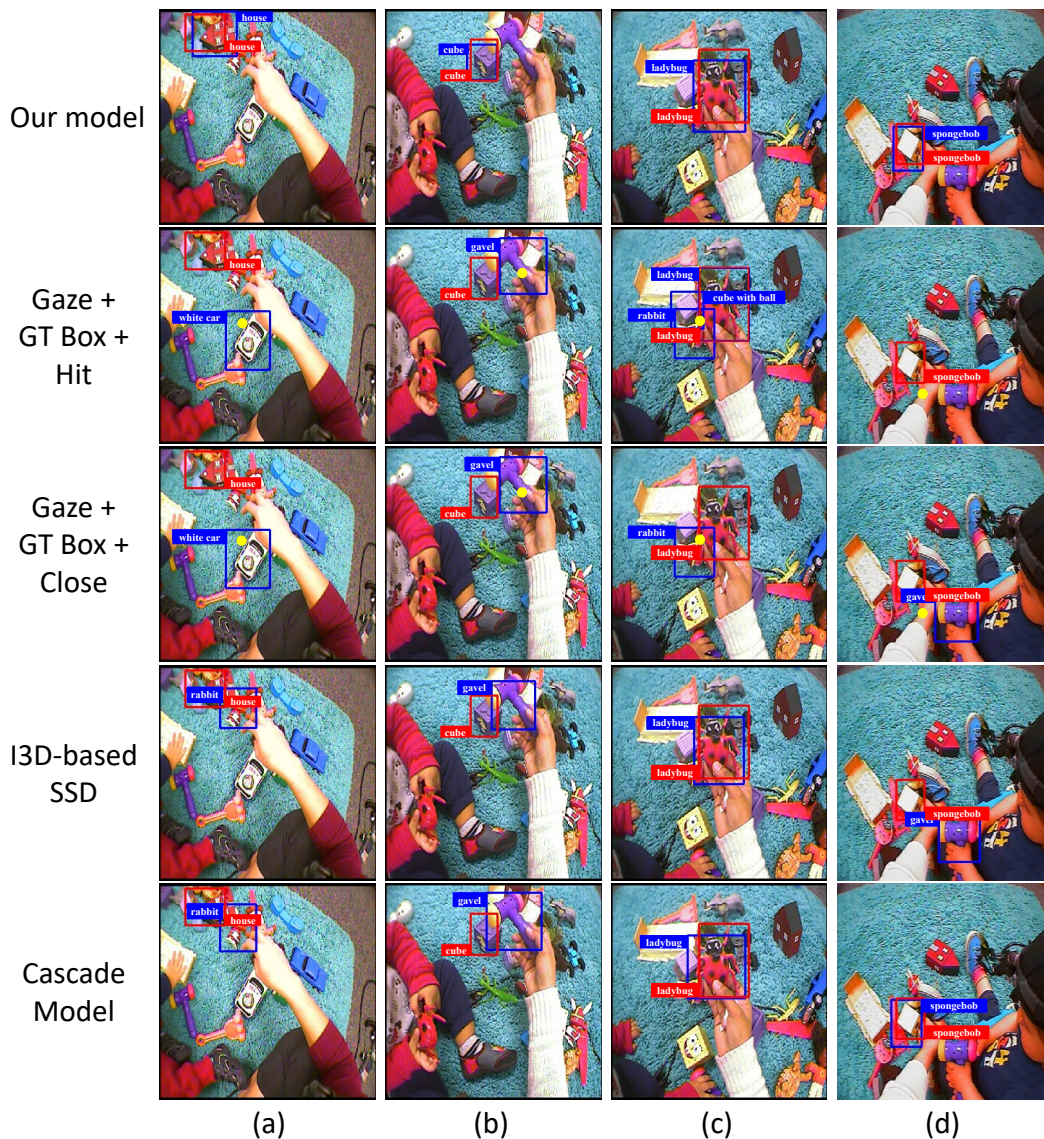

Figure 3: *Sample results of our Mr. Net and baselines on ATT dataset.* Detections are in blue, ground truth in red, and the predicted gaze of gaze-based methods in yellow.

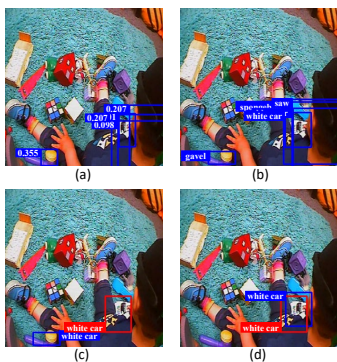

Figure 4: *Illustration of how parts of our model work.*

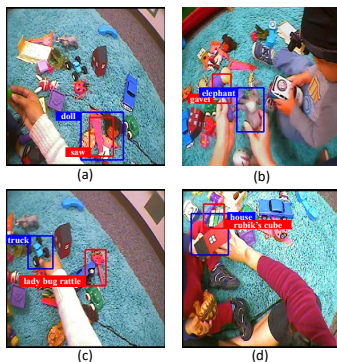

Figure 5: *Some failure cases of our model,* with detections in blue and ground truth in red.

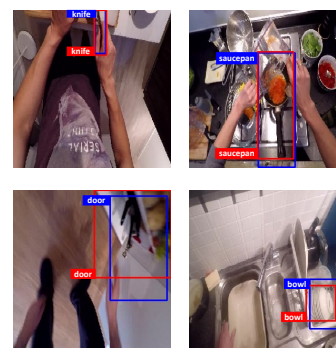

Figure 6: *Sample results of Mr. Net on Epic-Kitchens.*

| Model | $Acc_{0.5}$ ↑ | $Acc_{0.75}$ ↑ | $mAcc$ ↑ |
|---|---|---|---|
| Mr. Net | **71.34** | **38.26** | **39.04** |
| Gaze [68] + GT Boxes Hit | 26.46 | 26.46 | 26.46 |
| Gaze [68] + GT Boxes Closest | 36.81 | 36.81 | 36.81 |
| I3D [9]-bsaed SSD [40] | 67.43 | 37.90 | 37.22 |
| Cascade Model | 65.96 | 38.01 | 37.93 |

Table 3: *Results of online detection.*

| Method | $Acc_{0.5}$ ↑ | $Acc_{0.75}$ ↑ | $mAcc$ ↑ |
|---|---|---|---|
| Our Mr. Net | **57.18** | **31.00** | **31.20** |
| I3D [9]-based SSD [40] | 47.58 | 24.38 | 25.42 |
| Cascade Model | 51.20 | 28.18 | 28.36 |

Table 4: *Accuracies on the Epic-Kitchen dataset.*

directly, we observed a very poor result of $mAcc = 18.4\%$; adding the Self Validation Module back during testing yields a large gain to $mAcc = 25.1\%$.

**Alternative matching strategy for box attention prediction.** For the anchor box attention predictor, we perform experiments with different anchor matching strategies. When multi anchor matching is used, we do hard negative mining as suggested in [40] with the negative:positive ratio set to 3. The model with the multi anchor matching strategy achieves $mAcc = 44.27\%$, versus $mAcc = 44.78\%$ with one-best anchor matching. We tried other different negative:positive ratios ( *e.g.* 5, 10, 20) and still found the one best anchor matching strategy works better. This may be because we have an acceptable number of anchor boxes; once we set more anchor boxes, multi matching may work better.

**Object of interest class prediction.** We explore where to place the global object of interest class predictor. When we connect it to the temporal branch after the fused block 5, we obtain $mAcc = 44.78\%$; when placed after the conv block 8 at the end of the temporal branch, we achieve $mAcc = 43.69\%$. This implies that for detecting the object of interest among others, a higher spatial resolution of the feature map is helpful.

### 4.4 Online Detection

Our model can be easily modified to do online detection, in which only previous frames are available. We modified the model to detect the object of interest in the last frame of a given sequence. As shown in Table 3, except for the Gaze + GT boxes model, all other models suffer from dropping $Acc$ scores, indicating that online detection is more difficult. However, since the gaze prediction model that we use [68] is trained to predict eye gaze in each frame of the video sequence and thus works for both online and offline tasks, its performance remains stable.

### 4.5 Results on Epic-Kitchen Dataset

We show the generalizability of our model by performing experiments on Epic-Kitchens [13]. Results by applying our model as well as the I3D-based SSD model and the cascade model on this dataset are shown in Table 4. On this dataset, the $Acc_{0.5}$ of the Cascade model is higher than that of the I3D + SSD model. The reason may be that objects are sparser in this dataset and thus poorly-predicted boxes will be less likely to lead to wrong classification. Sample results are shown in Figure 6.

## 5 Conclusion

We considered the problem of detecting attended object in cluttered first-person views. We proposed a novel unified model with a Self Validation Module to leverage the visual consistency of human vision system. The module jointly optimizes the class and the attention estimates as self validation. Experiments on two public datasets show our model outperforms other state-of-the-art methods by a large margin.

## 6 Acknowledgements

This work was supported in part by the National Science Foundation (CAREER IIS-1253549), the National Institutes of Health (R01 HD074601, R01 HD093792), NVidia, Google, and the IU Office of the Vice Provost for Research, the College of Arts and Sciences, and the School of Informatics, Computing, and Engineering through the Emerging Areas of Research Project "Learning: Brains, Machines, and Children."

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
