[Supplementary Material]

# 7 Supplementary Material

## 7.1 The architecture of the Cogged Spatial-Temporal Module

Figure 7: *The architecture of the Cogged Spatial-Temporal Module.* The number below each component indicates its output dimension. $c$ is the number of class. All fusion is performed by element-wise sum. When trained without being followed by the Self Validation Module, before computing $L_{globalclass}$, $L_{boxclass}$ and $L_{attn}$, Softmax is applied (the attention prediction is first flattened to be a 8732-d vector)

Figure 8: *Visualizations of (a) Our model, (b) Gaze-based model, (c) Cascade Model, and (d) I3D-backboned SSD.* Note that in our experiments of Gaze + Box model, we directly use ground truth bounding boxes for each object instead of results from an object detector. The box regression head is omitted for simplicity.

## 7.2 Hand based model settings

We train two object-in-hand detectors (for the left hand and the right hand respectively), using the ResNet-50 backbone, and one which-hand classifier with the I3D backbone to classify which hand holds the object of interest when the left hand and the right hand hold different objects. During testing, if only one object-in-hand detector predicts object in hand or both hands hold the same object, we accept the prediction as the object of interest and it is combined with the ground truth bounding box as the final output. Otherwise we apply the which-hand classifier to decide which object to take.

We obtain testing accuracy of $86.28\%$, $88.61\%$ and $90.80\%$ for the object-in-left-hand detector, the object-in-right-hand-detector and the object-in-which-hand classifier respectively.

To further strengthen the baseline, we directly use the ground truth of objects in hands and have 4 more settings: (1) Right handed model, which uses the ground truth object in hands labels, and when two hands hold different objects, it always favours the right one; (2) Left handed model, which is the same as (1) but always favours the left hand; (3) Model with object-in-hand ground truth and which-hand classifier, which will apply the which-hand classifier to decide which object to take when two hands hold different objects; (4) Either handed model, which uses the ground truth object-in-hand labels, and when two hands hold different objects, the model always take the one resulting in higher $mAcc$ as the prediction. Note that (4) depicts the best performance which hand-based methods can possibly achieve in theory as it uses all of the ground truth.

Figure 9: *More qualitative results of our model on the ATT dataset.*