[Reviews · NeurIPS 2019]

Reviewer 1



This work seems interesting, but I found the paper very confusing. Here are some comments and questions: - The Introduction (and Figure 1 in particular) makes one think the paper addresses the join problem of predicting the location (bounding box) of the object of interest and recognizing the object. However, the results (as far as I understood) only test the mAP, meaning the accuracy on predicting the bounding box of the object of interest. Is the class prediction also tested quantitatively? - The model seems interesting, but I still didn't understand completely the purpose of the Temporal Gear Branch. While the Spatial Gear Branch acts as an object detector, what is the expected contribution of the Temporal Gear Branch? The text seems to suggest that it acts as an attention module, but I don't see exactly how (particularly when the Self-Validation Module is removed -in the ablation studies-). - In the qualitative results (Figure 3), it would be useful to see also how the baselines perform, for comparison. I assume authors made some visualizations, maybe they can discuss about their observations.

Reviewer 2



== Originality == The main contribution of the paper is a novel idea (to my knowledge). Speficially, the self-validation module proposed to combine and update location and class labels is very interesting, and is more original than the typical combination of different task-specific architectures to get a new result. I am not an expert on the field, but the related work seems adequate. == Quality == The model proposed in the paper seems to be technically sound, and each part has a role which is well-justified and properly explained. The choices for the "backbones" of the different architecture parts (e.g., SSD300 for the spatial object detector part) all make sense, and the loss functions are all well justified. Something was not clear: does the validation module have any trainable parameters? The experiments mostly show the approach achieves a significant improvement over several baseline algorithms of varying complexity. It is interesting to note that the SSD model baseline performs comparably to Mr Net, even though it is a much simpler model, and the paper should have discussed this point in more depth. Table 2 also has some interesting results. Specifically, the model that only uses RGB streams shows performance which is quite comparable to the full model, and is even superior for AP0.5. Though this is discussed in lins 265-272, the question of whether the optical flow input is providing any value is not really addressed. The online detection and EK dataset results are both very interesting and impressive, and provide good evidence to support the claims of the paper. Finally, I would be interested in seeing more qualitative results. Specifically, a discussion (including perhaps some images) about what types of mistakes this model typically makes when predicting, especially when compared to the baseline approaches, would be very useful. == Clarity == For the most part, the paper is well-written and easy to follow. A few relatively minor points: * please fix grammatical errors (e.g., plural vs. singular nouns) * the paper does not mention how many parameters the model has to learn * as asked above, it is not obvious whether the self-validation module has any parameters

Reviewer 3



- The paper is well written and the model details are clear for the reader. - The model is evaluated in two well-known datasets and compare to a handful of relevant baselines. - Results suggest the model improves over previous state of the art. The models seems to advance the state-of-the-art. - I believe it would be interesting for the reader to better understand the effect of the time window on the final computation. - I believe the reader would benefit from more failures examples in the paper. The examples shown are all success examples.

[Author Response · NeurIPS 2019]

Thanks for the insightful and helpful reviews, which will significantly improve our paper. Below, we refer to our Self Validation Module as SVMo. R1, R2, R3 indicate to whom the concern belongs. All figures can be zoomed in for better view. Ground truth is in red, predictions are in blue, and predicted eye gaze point of the gaze-based model is in green.

**Novelty [R2]:** First, we propose a novel, effective, flexible (lines 43-48) and robust method of mutual self validation
5 to model human foveated vision, inspired by both cognitive science and computer vision. Blurring has been used to simulate foveation, but this greatly reduces global context. Our SVMo does not require blurring yet can efficiently use the consistency of our vision system. SVMo bridges global and local context both spatially (*e.g.*, whole frame *vs* anchor box) and temporally (*e.g.*, video *vs* single frame). Second, we systematically study object-level attention and show that joint prediction of attended object class and location benefits each other. Existing work on object attention [5, 6] only
10 predicts location, not class. Other contributions include exhaustive experiments which may be useful for future studies.

**Qualitative results [R1,R2,R3] comparing our model with baselines** are in Fig. 1. **Regular gaze prediction models fail [R3]** in (c)&(d) due to the discrepancy between what and where (note we use ground truth bounding boxes in the gaze-based model, so failure is entirely caused by eye gaze prediction). In (c), the predicted gaze falls on the intersection of 3 objects, slightly closer to the center of the rabbit. In (d), eye gaze doesn't fall on any object. The
15 discrepancy makes gaze-based methods not robust to small shifts in gaze estimation. A more unified model (I3D-based SSD, the cascade model, and our model) thus achieves better performance. In (a) & (b), we show where our model outperforms I3D-based SSD and the cascade model. Because we use SVMo to inject consistency, Mr. Net performs better when many objects including the object of interest are densely close to each other.

**Sample failure cases of our model [R1,R2,R3]** in Fig.2: (a) heavy occlusion, (b) ambiguity of which held object is
20 attended, (c) the model favors the object that is reached for, and (d) an extremely difficult case where parent's reach is occluded by an object held by the child. We looked for examples where our model fails while baselines are correct, but have found none; when our model fails, so do the baselines.

**The effect of each part [R1,R3]**, including **the role of the temporal branch [R1]:** see Fig. 3 for an example. (a) shows the intermediate anchor attention score $A \in R^{a \times 1}$ from the temporal branch, visualized as the top 5 attended
25 anchors with attention scores. It is anchor-level attention and no box offsets are predicted here. Besides $A$, the temporal branch also predicts a latent global object of interest class score $C_{global} \in R^{1 \times c}$. (b) shows visualizations of the predicted anchor offsets $O \in R^{a \times 4}$ and box class score $C_{box} \in R^{a \times c}$ from the spatial branch (only of the top 5 attended anchors). (c) combines outputs from both branches; this is the final prediction of the model trained with SVMo but tested without it in the ablation studies. The predicted class is obtained from $C_{global}$ and we combine $A$ and $O$ to get the location. In
30 this example, the class prediction is correct but not the location. (d) shows prediction of our full model. By applying SVMo, the full model correctly predicts location and class. We also did two new experiments: (1) train with SVMo, test with only the what→where validation (the first step of our SVMo); (2) train without SVMo, test with only the what→where validation. We got $43.88\%$ mAP and $37.87\%$ mAP respectively, indicating both steps of the SVMo help.

**Intermediate representations [R3]** are qualitatively
35 visualized in Fig.3 and discussed above. In our ablation studies, the model trained with SVMo but tested without it gives reasonable $mAP$, quantitatively suggesting these representations are highly meaningful.

**Evaluating class labels [R1]:** Yes, the class is also
40 evaluated; a prediction is correct if it both (a) has the correct class and (b) the IoU between the estimated and the ground truth boxes is above a threshold.

**Trainable parameters [R2]:** The model has $64M$ parameters. SVMo has no trainable parameters, mak-
45 ing it very flexible and can be added to training or testing anytime. Also, we can only use half of the validation module (lines 31-33) and still get improvement. It is even possible to stack multiple SVMo's.

**I3D-based SSD *vs* our model [R2]:** The baseline
50 SSD uses the very strong feature extractor of Inception I3D, the same as our temporal branch. Mr. Net still achieves $10\%$ improvement over I3D-based SSD on ATT, and even greater improvement ($23\%$) on Epic-Kitchens, suggesting our model is more competitive on challenging datasets.

**1- *vs* 2-stream results [R2]:** It's interesting that RGB stream alone achieves higher $AP_{0.5}$ than our full model but lower $AP_{0.75}$ and $mAP$, suggesting flow leads to more high-quality detections. We hypothesize that optical flow helps better
55 identify where is attended: *e.g.*, hand movement may indicate reaching, which is correlated with attention.

**Time window [R3]:** Thanks. We will do this for the camera ready.

**Grammar and typos [R2].** Thanks. A native English speaker will proofread the final draft.

Figure 1: *Results from Mr. Net and baselines.* For Gaze+GT Box+Hit, there can be zero or more predictions.

Figure 2: *Some failure cases of our model.*

Figure 3: *Example of how parts of our model work.*

[Meta-Review · NeurIPS 2019]

All reviewers recommend accept. The authors are suggested to make the title less sensational, in particular the word Mindreader.